# First results from the equatorial geomagnetic station at Entoto Observatory and Research Center

Amoré Nel<sup>1,2</sup>, Nigussie Giday<sup>3</sup>, Marcos Da Silva<sup>4</sup>, Daniel Chekole<sup>3</sup>, Jürgen Matzka<sup>4</sup>, Ziyaad Isaacs<sup>1</sup>, Oliver Bronkala<sup>4</sup>, and Lamessa Mogasa<sup>3</sup>

Correspondence: Amoré Nel (anel@sansa.org.za)

- Abstract. This paper presents the initial results from the newly deployed Entoto Magnetometer Station near Addis Ababa,
- Ethiopia, a collaborative project involving the South African National Space Agency (SANSA), the Space Science and Geospa-
- tial Institute (SSGI) in Ethiopia, and the German Centre for Geosciences (GFZ). The station, equipped with a LEMI-025 flux-
- gate magnetometer and a GSM-90 Overhauser sensor, aims to monitor geomagnetic field variations and enhance space weather
- research in the African sector. This deployment is a significant step in SANSA's efforts to establish a comprehensive geomag-
- netic network across Africa, contributing to global space weather models. This is of particular importance, as the Entoto station
- is, to our knowledge, the only currently operational magnetic station near the dip equator in the African region, positioning
- the Entoto Observatory and Research Center at SSGI as a key contributor to regional and global geomagnetic research. Early

observations show a good characterization of geomagnetic disturbances, with observed field changes aligning closely with the

- Dst index variations, which has important implications for space weather forecasting. The station also generates local K-index
- data for this region, providing valuable insights into ionospheric variability and its effects on technological systems. This paper
- details the station's setup, data processing methodologies, and initial scientific results, laying the foundation for future research
- and collaboration in this critical area of space science.

## 15 1 Introduction

10

- The Earth's magnetic field plays a crucial role in shielding the planet from solar radiation and influencing space weather
- (Kamide, 2001; Kotzé et al., 2015). Generated primarily by the geodynamo in the Earth's liquid core, this field extends from
- the core through the planet's surface into space, where it interacts with solar wind and cosmic radiation. Other contributing
- sources include electrical currents in the ionosphere and magnetosphere, magnetized crustal rocks, and induced currents in the
- mantle and oceans. These internal and external sources collectively shape the geomagnetic field. Secular variation, the slow
- temporal change in the magnetic field, provides insight into the geodynamo's behavior, making long-term monitoring essential
- for understanding natural processes and the technological impacts of space weather (Nel et al., 2024).

<sup>&</sup>lt;sup>1</sup>South African National Space Agency (SANSA), South Africa.

<sup>&</sup>lt;sup>2</sup>Center for Space Research, North-West University, Potchefstroom, 2522, South Africa.

<sup>&</sup>lt;sup>3</sup>Department of Space and Planetary Science, Space Science and Geospatial Institute (SSGI), Addis Ababa, Ethiopia.

<sup>&</sup>lt;sup>4</sup>GFZ German Research Centre for Geosciences, Potsdam, Germany.

Ground-based geomagnetic observatories are vital for monitoring these variations. High-quality data from observatories have 23 been instrumental in studying secular variation and disturbances such as geomagnetic storms (Matzka et al., 2010; Nel and 24 Kotzé, 2024). Despite their importance, the global network of observatories faces challenges, particularly in under-monitored 25 26 regions like Africa and the Southern Hemisphere (Giday et al., 2020; Yizengaw and Moldwin, 2009). The scarcity of observations in these areas underscores the need for new stations, especially near the magnetic equator, where space weather strongly 27 28 influences the ionosphere (Mungufeni et al., 2018; Macmillan, 2007). The establishment of the Entoto Magnetometer Station in Ethiopia addresses this gap. Located near the magnetic equator, the station is part of SANSA's broader efforts to improve re-29 30 gional space weather forecasting. This collaborative initiative between SANSA, Ethiopia's SSGI, and Germany's GFZ, aims to 31 generate high-quality geomagnetic data to fill critical gaps in the global network. By capturing geomagnetic variations specific to the African sector, the Entoto station enhances both regional and global space weather models, contributing to improved 33 predictions of geomagnetic storms and their effects on technological systems, including communication networks and power grids (Matzka et al., 2010). 34

Several research fields are essential for understanding geomagnetic phenomena, particularly near the magnetic equator. Among these are studies of the equatorial ionosphere, equatorial plasma bubbles (Giday et al., 2020), geomagnetic storms, solar quiet (Sq) variations, the Equatorial Electrojet (EEJ), and the counter-equatorial electrojet (Habarulema et al., 2019). The EEJ, in particular, is of significant interest as it represents a concentrated eastward electric current superimposed on the global Sq current system. The relationship between the global Sq current system and the EEJ remains a subject of active research and ongoing debate (Yamazaki and Maute, 2016). Understanding the interactions between these currents is critical for improving models of ionospheric conductivity, geomagnetic variations, and space weather impacts on equatorial regions.

Despite the importance of such studies, Africa faces a persistent challenge due to the lack of active magnetometer stations along the magnetic equator. Currently, there are no operational magnetometer stations in the region with the necessary capabilities to effectively study Sq variations and the EEJ. Existing networks, such as the MAGnetic Data Acquisition System (MAGDAS) and the International Real-time Magnetic Observatory Network (INTERMAGNET), include a limited number of stations in Africa, but many are either inactive or positioned at latitudes too far from the equator to capture equatorial phenomena accurately.

For instance, there are two INTERMAGNET observatories in northern Africa: one in Ethiopia, one in Mbour, Senegal (Operated by IPGP), and one in Tamanrasset, Algeria (Operated by IPGP and CRAAG). However, both the Ethiopian and Senegalese stations have ceased recording data, and the Algerian station is located too far from the magnetic equator to be useful for equatorial studies. Similarly, the MAGDAS and African Meridian B-Field Education and Research (AMBER) networks, which previously had several stations along the magnetic equator, have experienced prolonged inactivity. As a result, Ethiopia currently lacks a functioning pair of low-latitude and equatorial stations, both of which are crucial for accurately studying EEJ dynamics and Sq variations.

Figure 1. Magnetic gradient survey of the Entoto Observatory area  $(9^{\circ}06^{'}33.6^{"}N, 38^{\circ}48^{'}24.1^{"}E)$ . The selected station location (marked) lies outside the highest-gradient region. Although peak gradients in the NW sector exceed 200 nT/m, the final site was chosen for minimal interference and is suitable for a variometer station intended for space weather monitoring and absolute observatory criteria (e.g. 

**Figure 2.** Magnetometer Setup: The magnetometer is housed in a protective structure designed to withstand environmental challenges, including potential flooding, and to maintain temperature stability. Data is logged continuously and transferred to the respective institutes for analysis and storage.

dynamics and their impact on communication, navigation, and power systems. This will improve the monitoring and prediction of space weather phenomena in Africa, especially in regions historically underrepresented in existing networks (Uemoto et al., 2010).

The local K-index, derived from ground-based measurements, is an essential tool for assessing local disturbance monitoring and prediction. Numerous space weather prediction centers have started measuring their local K-index using local magnetometer station data, in order to do regional forecasting: Unlike the planetary Kp-index, which averages data from multiple observatories worldwide, the local K-index provides region-specific insights into geomagnetic activity. For stations located near the magnetic equator, such as Entoto, the K-index must account for the unique influence of the equatorial electrojet, which can distort the magnetic field measurements used to compute the index. Recent studies, such as those conducted at the Phuket station in Thailand, have demonstrated the importance of selecting appropriate calibration values for generating accurate local K-indices in equatorial regions (Hamid et al., 2014; Myint et al., 2022). One of the key challenges in computing the local K-index near the equator is determining the lower limit (L<sub>9</sub>, which is defined as the lower K = 9 threshold value) for the K-index scale, which varies with geomagnetic latitude. Equatorial stations require higher L<sub>9</sub> values due to the strong influence of the EEJ, which can otherwise lead to an overestimation of geomagnetic disturbances. At the Entoto station, the analysis of the local K-index will provide critical insights into the day-to-day variability of the EEJ and its impact on geomagnetic activity.

- By generating accurate local K-indices, the station will enhance regional space weather prediction capabilities, particularly in
- terms of forecasting geomagnetic storms that could disrupt communication systems and power grids. Additionally, the data
- collected at Entoto will contribute to a broader understanding of how geomagnetic disturbances evolve in the African sector,
- filling a critical gap in the global geomagnetic observation network (Menvielle et al., 1995).
- The establishment of the Entoto magnetometer station marks a significant milestone in the advancement of geomagnetic re-
- search in Africa (Hamid et al., 2014; Myint et al., 2022; Mungufeni et al., 2018). By providing high-quality data on the local
- K-index and geomagnetic field variations, the station will play a pivotal role in improving space weather prediction capabilities
- both regionally and globally. The collaboration between SANSA, the SSGI, and the GFZ highlights the importance of regional
- efforts in addressing the global challenges posed by space weather.

### 2 Site Selection and Instrumentation

- A geomagnetic station is where the geomagnetic field vector is recorded continuously over a long period of time (Matzka et al.,
- 2010). Ideally, the site should be free from any static disturbances caused by local anomalies, as well as far away from human
- traffic that could cause temporal disturbances.
- The Entoto Observatory, established in 2014 as one of the facilities of the then Ethiopian Space Science and Technology
- Institute (now the Space Science and Geospatial Institute (SSGI)) represents a key step in advancing the nation's space science
- research. Funded by the Ethiopian government, along with support from universities, international partners, and private donors,
- the observatory offers the necessary infrastructure for research in space science, geospatial technologies, and astronomy.
- Located at the highest point in Addis Ababa, at an elevation of 3200 metres above sea level, it reflects SSGI's commitment
- to research and positions Ethiopia as a leader in regional and global space science. The Entoto Observatory is situated approxi-
- mately 15 km northeast of Addis Ababa. Its location away from local settlements and adjacent to Entoto National Park ensures
- minimal interference from local magnetic noise.
- A magnetic gradient survey was conducted (see Figure 1) and a deployment site was identified within the original section
- of the Entoto Observatory in the northwestern sector (9°06′33.6″N,38°48′24.1″E). It is about 20 meters from the western
- perimeter fence and 15 meters from an abandoned hut.
- The magnetic gradient survey (Figure 1) revealed localised regions of elevated gradients, with values exceeding 10 nT/m
- in the northwestern sector of the original Entoto site. While this surpasses the 1 nT/m threshold typically recommended for
- absolute geomagnetic observatories (Jankowski and Sucksdorf, 1996), that criterion is specifically intended for baseline quality
- installations contributing to the global magnetic reference network. In contrast, the Entoto deployment is not intended to
- operate as an INTERMAGNET-grade observatory, but rather as a variometer station focused on monitoring relative magnetic
- field variations for space weather research. In this context, moderate gradients, although not ideal, are acceptable, as variometer
- measurements are generally robust to static field anomalies.
- The selected deployment site was carefully positioned just outside the steepest gradient zone, where *in situ* testing confirmed
- improved magnetic cleanliness. The primary scientific objective is to capture dynamic ionospheric and magnetospheric field

perturbations, particularly those associated with phenomena such as the Equatorial Electrojet (EEJ), for which relative stability is more critical than absolute accuracy. Furthermore, future infrastructure plans include relocating the station to the western section of the Entoto campus, where the construction of new perimeter fencing and expanded facilities will allow for a 100 m buffer from any surrounding structures, significantly improving the site's magnetic environment and long term suitability.

The protective structure for the proposed magnetometer station is based on the design of SANSA's protective structure at our INTERMAGNET observatory in Keetmanshoop (KMH) (Korte et al., 2009). Some adjustments were made to this structure, for example, to compensate for potential flooding which could occur in Addis Ababa. The original structure was designed to be buried underground, the new structure will be above ground, with ample area underneath the floor for sudden water flow. Better venting pipes were designed for temperature stability.

# 2.1 Temperature stability and correction

136137

Fluxgate magnetometers are thermally sensitive, and venting pipes alone do not ensure long-term stability. To quantify the thermal environment at Entoto, we analysed the co-located sensor (T1) and electronics (T2) temperatures. Daily ranges were modest, with median values of  $\sim 3.5^{\circ}$ C (95th percentile 4.9°C) for T1 and  $\sim 4.6^{\circ}$ C (95th percentile 7.2°C) for T2 (Table 1). This corresponds to an estimated median peak-to-peak effect of only  $\sim 2.6$  nT on the H component. A representative average daily profile is shown in Figure 3, illustrating the moderated diurnal cycle inside the enclosure.

**Table 1.** Summary of daily temperature ranges inside the Entoto enclosure over the analysed period. Values are in °C.

| Quantity               | Median | Mean | 95th Percentile |
|------------------------|--------|------|-----------------|
| T1 (Sensor) Range      | 3.5    | 3.6  | 4.9             |
| T2 (Electronics) Range | 4.6    | 4.8  | 7.2             |

To mitigate residual thermal effects, we derived temperature coefficients using co-located sensors and quiet-day analysis. The H component showed a coefficient of -0.733 nT  $^{\circ}$ C<sup>-1</sup>, while HE and Z contributions were negligible. As illustrated in Figure 4, applying this coefficient to H reduces the small daily variation in  $\Delta F$  (about 2 nT on a quiet day) and lowers its correlation with temperature. The effect is subtle, and in some cases the corrected values show slightly more scatter, which likely reflects additional baseline adjustments in the calibration workflow rather than the temperature correction itself.

Importantly, the correction does not degrade the data quality, and the overall  $\Delta F$  variation remains dominated by geophysical signal rather than thermal artefacts.

Even after correction,  $\Delta F$  offsets of  $\sim$ 6–8 nT remain, reflecting uncertainties in baseline estimation and sensor differences rather than thermal effects. We therefore consider the enclosure and correction procedure adequate for variometer-grade operation, while acknowledging that this does not reach observatory-grade stability.

Figure 3. Average daily temperature profile inside the Entoto station enclosure, derived from all days of the last months . sec data (09-08-2025 to 09-09-2025). Sensor temperature (T1) and electronics temperature (T2) both show modest diurnal variations (median daily ranges of  $\sim 3.5^{\circ}$ C and  $\sim 4.6^{\circ}$ C, respectively), confirming that the enclosure moderates the thermal environment.

Figure 4. Comparison of  $\Delta F$  uncorrected (yellow) and corrected (blue) applying the H-component temperature correction ( $-0.733 \text{ nT} \,^{\circ}\text{C}^{-1}$ ) on a representative quiet day (14 August 2025). The correction slightly reduces the small diurnal variation ( $\sim$ 2 nT) and lowers correlation with temperature, consistent with the expected thermal sensitivity of fluxgate sensors.

#### 2.2 Data logging and instrumentation

In partnership with the GFZ, a converter program for the LEMI magnetometer data recorder has been written. The station currently does not include dedicated lightning protection due to the secure facility layout and non-grounded sensor cables. 141 However, lightning protection measures are under consideration for future upgrades. The LEMI-025 is connected via RS232 142 to a Linux PC and the binary data is converted to readable ASCII format. The bias signals and DAC values will be monitored 143 144 and checked for any anomalies. 145 The Entoto magnetometer station is equipped with a GSM-90 Overhauser sensor and a LEMI-025 fluxgate magnetometer. 146 The GSM-90 is a portable Overhauser magnetometer that measures the total magnetic field strength with high accuracy. Its 147 application varies, but for this project it's implemented for long term use and will measure the scalar signal of the local geomagnetic field. The LEMI-025 is a sensitive 3-axis fluxgate magnetometer (FGM) that measures the three components of 148 the geomagnetic field (thus providing directional information) and records temporal variations at a resolution of 1 second. The 149 LEMI-025 was placed on a magnetometer pillar within the protective structure and the GSM-90 Overhauser a few metres from 150 the structure (see Figure 2) and further adjustments and testing will ensue, e.g., looking for reasonable variations, timestamps, 151 152 adjusting the placement of the instruments, checking for disturbances in the serial line etc. With this dual magnetometer setup one can obtain both the total strength and direction of the local magnetic field, and enables a more complete understanding 153 154 of geomagnetic variations. Although the GSM-90 is highly accurate, it has a lower sampling rate than the LEMI-025, thus combining these two sensors it provides both precision and detailed time series data. When studying space weather phenomena, 155 156 both the intensity and the direction of the geomagnetic field changes are crucial. Using both instruments at geomagnetic observatories helps to monitor geomagnetic storms, substorms, and ionospheric phenomena effectively. 157 158 The sensor house comprises a reinforced structure to protect against environmental challenges. The magnetometer sensor is connected to the control room via a cable, ensuring stable power supply through a power system. The magnetic field is 159 represented by its three components, North (X), East (Y), and vertical (Z), measured with respect to the local geographic 160 coordinate system, with Z being positive downward (Denardini et al., 2015). These components are digitized and logged, 161 enabling high temporal resolution analysis for space weather monitoring.

#### 3 Methodology 163

The Entoto magnetometer station records variations in the geomagnetic field along the X, Y, and Z components with high temporal resolution. These measurements enable the identification of geomagnetic activity patterns, including responses to solar 165 wind disturbances and geomagnetic storms. The collected data undergoes a series of processing steps to extract meaningful 166 geophysical signals while mitigating noise and long-term trends. The 1-minute values were computed from 1-second magne-167 tometer data using a centered arithmetic mean, following IAGA and INTERMAGNET standards, where each value represents 168 169 the average of 60 samples centered on the target minute. Initial testing of the instrumentation was conducted at SANSA's facilities in South Africa before deployment in Ethiopia to ensure proper calibration and data integrity. 170

Several non-geomagnetic noise sources can influence the magnetometer signal at Entoto, including temperature-induced drift and anthropogenic electromagnetic interference (e.g., from nearby infrastructure or power lines). Among these, temperature effects are corrected through post-processing using a derived temperature coefficient for the H component. However, other noise sources are more difficult to remove without risking the attenuation of the signal of interest: EEJ and Sq signals primarily occupy ultra-low frequency (ULF) ranges, with dominant diurnal and semi-diurnal components below 1 mHz (Yamazaki and Maute, 2016). While these frequencies are generally well-separated from anthropogenic sources such as power-line harmonics (50–60 Hz) (Constable and Constable, 2023), low-frequency electronic noise can introduce artefacts in the sub-mHz band. Additionally, poor shielding or digital aliasing may fold high-frequency noise into lower frequencies, complicating the reliable extraction of ionospheric signals. Therefore, we adopt a conservative filtering strategy: while a high-pass filter and quiet-day subtraction are applied to isolate short-term variations, no aggressive denoising is performed. This trade-off preserves the geophysical integrity of the data, especially in studies focused on daily-scale variability and ionospheric current systems.

To isolate external geomagnetic variations, the main field, which mainly originates from the Earth's core, is subtracted using the CHAOS 8.2 model (Kloss et al., 2025). This model estimates the internal geomagnetic field at the station's coordinates during analysis and is limited to 2025.1 to avoid extrapolation. These estimated values are subtracted from the observed data to obtain the corrected geomagnetic field components, removing long-term variations such as secular changes and leaving only the external field perturbations. This step ensures that local geomagnetic fluctuations are not influenced by global-scale internal variations.

To study short-term geomagnetic variations, the residual field components undergo further processing. A Butterworth high-pass filter with a cutoff period of approximately 72 hours is applied to remove long-term variations. To account for solar quiet variations, a quiet-day mean for each month is computed and subtracted from the dataset. Finally, a daily running mean is subtracted from the data to eliminate long-period fluctuations. This process isolates daily geomagnetic variations primarily influenced by ionospheric and magnetospheric currents, allowing for a more precise analysis of space weather effects.

To provide a broader context for local geomagnetic disturbances, the planetary Ap index is incorporated into the analysis. The maximum daily Ap index values are overlaid on the corrected geomagnetic field plots, with color-coded markers indicating active disturbance conditions for Ap values greater than or equal to 19 and geomagnetic storm conditions for Ap values exceeding 36. The integration of Ap index data allows for direct comparisons between local fluctuations and global geomagnetic activity. This step is essential for correlating regional space weather phenomena with larger-scale geomagnetic disturbances that could impact communication systems and navigation infrastructure.

### 3.1 Estimating the local K index

The K-index quantifies geomagnetic disturbances on a quasi-logarithmic scale from 0 (quiet) to 9 (strong storms). The Entoto station's local K-index is computed using the open-source MagPy software (Stolle et al., 2018) following the Finnish Meteorological Institute (FMI) method, which is widely used in geomagnetic observatories. The International Association of Geomagnetism and Aeronomy (IAGA) has standardized four computer-based K-index algorithms that can be applied glob-

ally. These algorithms use different solar regular (SR) curve estimation techniques. Among them, the FMI method proposed by Sucksdorff et al. (1991) is considered the most reliable compared to manual hand-scaling techniques. The FMI method is widely used for K-index generation at various observatories and serves as a baseline method for evaluating newer approaches, such as the nowcast K-index used in space weather forecasting. The FMI method employs a linear elimination approach, using geomagnetic field data from three consecutive days to estimate the SR curve for a given day.

The calculation of the local k-index using the FMI method is dependent on a station specific constant, called  $L_9$ . This value represents the threshold geomagnetic range (in nanoTesla) that corresponds to the highest level of activity (K = 9) at that station.  $L_9$  is not a dynamic variable, but a fixed value that is determined empirically using long-term historical data, typically over several years or a full solar cycle. It reflects the station's geomagnetic latitude and local environmental conditions, such as the proximity to the local EEJ. It is usually analyzed using the statistical distribution of magnetic ranges to align with global K-index conventions.

While software like MagPy can compute K-indices in realtime, it requires a pre-defined  $L_9$  value to map the observed 3-hour range maxima to the appropriate K-level. For the Entoto station outside Addis Ababa, it is possible to use the historical  $L_9$  value from the now decommissioned INTERMAGNET station AAE, provided that the geomagnetic environment and signal processing are comparable.

Derivation of the K-index at the Entoto station begins by importing high-resolution time series raw data at minute intervals. A baseline correction is applied and filtering to remove non-geomagnetic noise. The geomagnetic time series is then segmented into 3-hour windows, and for each interval, the maximum and minimum values of the horizontal field components are determined to compute the fluctuation range. The thresholds account for the influence of the equatorial electrojet, which can introduce distortions to standard K-index computations. Based on the fluctuation range, MagPy assigns a K-index value between 0 and 9 to each 3-hour period. This approach ensures that the K-index calculated by MagPy is both precise and station-specific, capturing local geomagnetic disturbances that may affect technological systems or contribute to space weather research.

The next few subsections outline the procedures to isolate the equatorial electrojet (EEJ) and magnetospheric contributions from low-latitude ground-based geomagnetic observations. The approach is based on removing internal and large-scale external field components from the horisontal magnetic field using the CHAOS 8.2 model, and separating diurnal (ionospheric) and nocturnal (magnetospheric) variations.

# 231 3.2 Data Preparation

Geomagnetic data with one-minute resolution from the Entoto station was used to determine the horizontal magnetic field strength. The observed horizontal magnetic field strength,  $H_{\rm obs}$ , was computed from the orthogonal magnetic field components X and Y as:

$$H_{\text{obs}} = \sqrt{X^2 + Y^2}$$
. (1)

### 236 3.3 Internal Field Removal

- To account for the Earth's internal field, we used the CHAOS geomagnetic field model (Finlay et al., 2020) to estimate the
- internal horizontal field,  $H_{int}$ , at the station's geographic location and for each observation time. The residual horizontal field
- was obtained by subtracting the internal field:

$$H_{\rm res} = H_{\rm obs} - H_{\rm int}$$
. (2)

This residual field contains contributions from both ionospheric and magnetospheric sources.

# 242 3.4 Equatorial Electrojet Signal Extraction

- To isolate the ionospheric EEJ signal, we selected the  $H_{\rm res}$  values corresponding to local daytime hours, typically from 09:00
- to 15:00 LT, when EEJ activity is strongest (Onwumechili, 1997; Rangarajan et al., 2002). This subset of the residual field
- represents the unrefined EEJ signal.

# 246 3.5 Magnetospheric Contribution Isolation

- The nighttime portion of the residual field, spanning 18:00 to 06:00 LT, is assumed to be dominated by magnetospheric
- contributions due to the absence of significant ionospheric currents. This signal was extracted and analyzed alongside the
- global Dst index (Sugiura, 1964) to assess their correlation, which will be discussed in more detail in the Results section.
- To assess the capability of the Entoto station to detect space weather effects, we generated a time series comparing the local
- magnetospheric signal, extracted from nighttime residuals of the horizontal magnetic field, with the global Dst index. Then
- we selected geomagnetically quiet periods (Dst > -20) and disturbed periods (Dst < -50) in order to show superposed epoch
- plots of the EEJ signal, showing its diurnal variation averaged over each period. The first method will illustrate the station's
- sensitivity to global magnetospheric conditions, while the second will show how ionospheric currents (particularly the EEJ)
- behave during quiet and disturbed times. To support this visual analysis, we will also perform two quantitative comparisons:
- a Pearson correlation between the magnetospheric signal as measured at the station and Dst to measure the linear association,
- and a comparison of the mean daily maximum EEJ amplitude during quiet and storm conditions. These metrics provide a
- simple statistical measure of the station's physical response to varying levels of geomagnetic activity.

#### 259 4 Results and Discussion

- Figures 3 and 4 illustrate the thermal environment and its correction at Entoto. Figure 3 shows that both the sensor (T1) and elec-
- tronics (T2) temperatures exhibit only modest diurnal variations, while Figure 4 demonstrates that applying the H-component
- temperature coefficient (-0.733 nT  $^{\circ}$ C<sup>-1</sup>) slightly reduces the small daily variation in  $\Delta F$  and lowers its correlation with tem-
- perature. These results confirm that the enclosure provides adequate thermal moderation for variometer-grade operation, with
- post-processing correction acting as an additional safeguard.

**Figure 5.** Variations in the local geomagnetic field along the X (red line), Y (green line), and Z (blue line) components during November 2024. At the top of the graph the daily maximum Ap value is shown. The greyed out days refer to quiet times, yellow days are unsettled, and red is disturbed days.

**Figure 6.** Kp and local K indices derived from the Entoto station. Both figures show indices as measured during disturbed storm times, as seen in Figure 5.

The processed diurnal variations were visualized by plotting the residual components  $\Delta X, \Delta Y, \Delta Z$  against local time to observe the daily geomagnetic patterns. Plots were generated for each complete month since deployment until January 2025 as shown in Figures 5, 7, and 9. Additionally, solar activity indices, including the Ap index, were compared with the diurnal variations to evaluate the influence of solar activity on the geomagnetic field. Based on the Ap-index values, the above analyses show that the geomagnetic field variations were consistent with the planetary geomagnetic activity levels. During the night, the fields were generally stable for quiet days. When the geomagnetic activity was high, irregular variations were detected during both daytime and nighttime.

The comparison between the global planetary Kp index and the locally derived K index at the Entoto station consistently shows that the local K values are approximately one to two levels higher, even during geomagnetically quiet periods, shown for selected times in Figures 6 and 8. This discrepancy can largely be attributed to the station's location near the magnetic equator, where it is strongly influenced by the Equatorial Electrojet (EEJ). The EEJ introduces pronounced diurnal variations in the horizontal component of the geomagnetic field, which are not captured in the mid-latitude stations contributing to the Kp index (Forbes and Lindzen, 1981; Onwumechili, 1997). Consequently, the Entoto station records elevated field disturbances relative to global averages.

In the present study, a historical K9 threshold value of 242 nT, derived from the former AAE INTERMAGNET observatory in Addis Ababa (latitude 9.035° N, longitude 38.77° E), was used to calculate the local K index. However, the significance of this threshold is limited by the fact that the station was located at a geomagnetic latitude of less than 10°, where the EEJ strongly modulates magnetic variability and distorts comparisons with mid-latitude standards. As such, this K9 value may not adequately characterize the magnetospheric and ionospheric influences at low-latitude sites like Entoto. As part of future work, a dedicated and empirically determined L<sub>9</sub> value will be established for Entoto once a longer time series of high-quality data becomes available (Menvielle et al., 1995; Korte et al., 2018).

Figures 10, 11, and 12 show the measured magnetospheric signal of the station versus the global Dst index, the mean EEJ signal during quiet (Dst > -20), and disturbed (Dst 

**Figure 7.** Variations in the local geomagnetic field along the X (red line), Y (green line), and Z (blue line) components during December 2024. As described in Figure 5

Figure 8. Kp and local K indices derived from the Entoto station. Here Figures show indices during quiet times, as seen in Figure 7.

**Figure 9.** Variations in the local geomagnetic field along the X (red line), Y (green line), and Z (blue line) components during January 2025. As described in Figure 5

Figure 10. The measured magnetospheric signal of the station versus the global Dst index for date range 2024-10-18 to 2025-01-31

**Figure 11.** Mean EEJ amplitude for selected quiet geomagnetic days for the selected date range, based on the daily mean of 1-minute EEJ values. These values provide a baseline for comparison with disturbed-time EEJ enhancements.

**Figure 12.** Mean EEJ amplitude for selected geomagnetic storm days for the selected date range, computed as the daily mean of 1-minute EEJ values from the Entoto variometer data. This captures the net enhancement of the equatorial electrojet during disturbed conditions.

### 295 5 Conclusion

Variations in the local geomagnetic field along the X, Y, and Z components were plotted alongside the maximum daily Ap values. This was used to identify recurring patterns and assessing the impact of space weather over extended periods. This preliminary data from the Entoto Magnetometer Station shows clear variations in the geomagnetic field, particularly during periods of increased solar activity.

The local K-index for the Entoto station was estimated using the MagPy software and compared against the planetary K-index. These results are consistent with previous studies (Kotzé et al., 2015), which have demonstrated the value of the local K-index in assessing geomagnetic activity, particularly in regions with distinctive geomagnetic features. In the case of Entoto, situated near the magnetic equator, the influence of the equatorial electrojet (EEJ) plays a significant role. As expected, the local K-index was typically higher than the planetary K-index, both during quiet and disturbed periods. This discrepancy underscores the importance of generating a station-specific L9 (K9 lower limit) value once sufficient data has been collected. While this would improve the precision of the local K-index, determining an appropriate L9 value in equatorial regions is inherently challenging due to the dynamic and variable nature of the EEJ.

The gradient survey revealed localised crustal anomalies with values exceeding 10 nT/m in some areas. These elevated gradients are indicative of the magnetic properties of the underlying crustal rocks and do not, by themselves, imply enhanced electromagnetic induction. Induction effects, when present, arise from contrasts in subsurface electrical conductivity rather than from magnetic gradients. Both phenomena may coexist in complex geological settings and could introduce non-geomagnetic signals during storm times. However, since the Entoto Magnetometer Station operates as a variometer (rather than a full absolute observatory), its primary role is to capture temporal variations in the geomagnetic field. Any induction-related variability is therefore not expected to compromise the station's scientific objectives, which focus on monitoring dynamic current systems such as the Equatorial Electrojet (EEJ) and storm-time magnetospheric responses.

Nevertheless, to improve the long-term reliability and data quality of the station, a future relocation ("lift and shift") is planned. The Entoto Observatory is currently undergoing expansion on its western side, where new perimeter fencing is being erected for improved security. The expanded area will allow for the installation of magnetometer infrastructure with a minimum clearance of 100 m from nearby buildings, fences, or ferromagnetic structures, meeting the site criteria recommended for modern observatories. This future upgrade will further reduce the influence from local anomalies.

Despite these challenges, further analysis across several geomagnetic storms revealed prompt increases across all field components, confirming the station's responsiveness to magnetospheric disturbances. The performance of the local K-index, though influenced by EEJ effects, demonstrates Entoto's capability to provide meaningful space weather data, aligning with methods applied in other equatorial observatories (Myint et al., 2022). The combination of CHAOS 8.2 internal and external field corrections, high-pass filtering, and preliminary regional K-index estimation forms a framework for interpreting magnetic field variations at this station, and creates a foundation for future upgrades.

Initial results demonstrate the station's ability to detect and characterize geomagnetic disturbances, with observed field changes aligning well with Dst index variations. The correlation between magnetospheric signals and global disturbance indices suggests that the Entoto station is sensitive to space weather drivers, hence proves useful for regional monitoring.

As the only current equatorial station on the African continent, the Entoto Magnetometer Station plays a critical role in addressing current gaps in longitudinal geomagnetic data. Its location on the magnetic equator enables detailed observations of equatorial ionospheric phenomena, including the Equatorial Electrojet (EEJ) and Sq currents, which are poorly resolved by mid- and high-latitude stations.

The computation of a local K-index using a historical  $L_9$  threshold of 242 nT, based on the former AAE magnetic observatory 15 kilometres away from the newly deployed station, highlights both the value and limitations of such metrics in equatorial regions. While the local K-index was consistently higher than the global Kp, this discrepancy, also observed in other studies (Myint et al., 2022), underscores the need to determine a site-specific  $L_9$  value for Entoto. However, the day-to-day variability introduced by the EEJ makes this a challenging task.

The comparison between the station's magnetospheric signal and the Dst index, along with the observed differences in EEJ amplitude during quiet and disturbed periods, highlights the capability of the Entoto Magnetometer Station to capture both global and regional geomagnetic variations.

So despite some challenges, the observed responses to geomagnetic storms, the correlation with Dst, and the strong performance of the preliminary K-index estimation all confirm the Entoto station's potential to deliver accurate and valuable space weather measurements. These results mark an important milestone for geomagnetic monitoring in Africa and demonstrate the station's readiness to contribute to the global geomagnetic network.

### 6 Future Work

While the initial results have focused on geomagnetic field variations, future studies will expand to include a deeper analysis of ionospheric effects, such as the Equatorial Electrojet (EEJ) and Sq variations. These phenomena are of particular interest for understanding how space weather affects the African sector. It is also imperative to calculate a local L9 value. Additionally, there is potential for expanding the network of magnetometer stations across Africa to improve longitudinal data coverage. The deployment of the Entoto Magnetometer Station marks a significant milestone in geomagnetic monitoring in the African region, but we also acknowledge the valuable foundation laid by previous magnetometer deployment initiatives across the African continent, including the AMBER network, whose efforts have informed our current approach. Drawing from these lessons, the Entoto station is supported by a formal Memorandum of Understanding (MoU) between SANSA and SSGI, ensuring long-term institutional engagement. Key to this partnership is ongoing, transparent communication across research and technical teams, enabling shared responsibility in both instrument maintenance and data utilisation. This collaborative model avoids silo-ed operations and fosters local capacity development. To support this, SANSA and SSGI continue to jointly apply for travel and training grants to facilitate technical workshops and ensure long-term sustainability.

Our initial results show that the station is well-equipped to contribute valuable data to global space weather research. With future expansions and collaborations, this station will play an increasingly important role in understanding geomagnetic and ionospheric phenomena in this critical region.

# **Author Contribution**

361

- AN, NG, ZI, and DC were responsible for the deployment of the Entoto station. MS, OB, and JM contributed to the processing
- of raw data into IAGA format and supported the development and operation of the data transfer software. AN performed the
- data analysis and drafted the results and conclusions sections.

# **366 Competing Interests**

The authors declare that there are no competing interests.

# 368 Data Availability

- The Entoto Magnetometer Station is not part of the INTERMAGNET network; however, we recognise the scientific value of
- sharing data from this equatorial variometer station. As such, the Entoto Magnetometer Station data are available for non-
- commercial scientific use upon request to the corresponding author. As a condition of use, users must cite this publication to
- acknowledge the source of the data and the contributions of the SANSA-SSGI-GFZ collaboration. Future steps may include
- integration into broader African or global networks via open-data agreements.
- Acknowledgements. The authors would like to acknowledge **Leonhardt Roman** for developing and maintaining the MagPy software suite,
- which was instrumental in processing and analyzing magnetic field data for this study. We thank Chane Moges and Yekoye Tariku for
- their assistance during the deployment of the magnetometer. We also extend our sincere thanks to Emmanuel Nahayo at the South African
- National Space Agency (SANSA) for his support in assessing the preliminary geomagnetic measurements from the Entoto station. Also **Oliver**
- **Bronkala** of the GFZ who assisted in transferring the data to a dedicated server.
- We acknowledge the use of the CHAOS geomagnetic field model, developed and maintained by the Technical University of Denmark
- (DTU), which was used to remove field contributions from the observations. We further acknowledge the providers of geomagnetic indices:
- the **Dst index** supplied by the World Data Center for Geomagnetism, Kyoto, and the **Ap index** provided by the GFZ German Research Centre
- for Geosciences in Potsdam.
- The Entoto dual magnetometer station is operated with the support of the Entoto Observatory and Research Center, with technical
- collaboration from SANSA and GFZ Potsdam. We also acknowledge the INTERMAGNET network and its data standards, which served as a
- reference framework during quality assurance.
- This research made extensive use of open-source tools, including Python, pandas, matplotlib, and SciPy, and we gratefully
- acknowledge the open-science community for their contributions. The author(s) acknowledge the use of the AI language model ChatGPT for
- assistance with Python syntax correction and Grammarly for language refinement in the preparation of this manuscript.

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
