# Peer review of "First results from the equatorial geomagnetic station at Entoto Observatory and Research Center"

_EGUsphere, 2025_

## Referee Comment (RC1)

**Referee Report:** First results from the equatorial geomagnetic station at Entoto Observatory and Research Center by A Nel et al.

**Summary**:

This paper presents the initial results from the newly deployed Entoto Magnetometer Station near Addis Ababa, Ethiopia, a collaborative project involving the South African National Space Agency (SANSA), the Space Science and Geospatial Institute (SSGI) in Ethiopia, and the German Centre for Geosciences (GFZ).

**General Comment:**

This is a well-written manuscript that can be recommended for publication after some re-writing has been done to address certain questions and concerns.

**Specific Comments and Questions**

1. The 'Entoto Observatory' can at best be called a 'Variometer Station' because it does not qualify as a magnetic observatory in the true sense of the word as no absolute measurements are being done to determine baseline values. In the manuscript the authors use both 'station' and 'observatory' which can confuse the reader of this paper.

2. In line 7 the authors write 'ENTOTO', while in the manuscript title it is written as 'Entoto'. Consistency is advised.

3. Figure 1 reveals that substantial magnetic gradients exist at the site of the station, reaching 240 nT/m in the North-West corner. The effect of this is that the area is not clean as a 10nT/m gradient is normally the criterium for a clean site. It is recommended that the position of the station be shown in Figure 1 to give the reader a better appreciation of its location. These large gradients, indicative of magnetised rocks under the surface, can unfortunately lead to substantial induction effects on the magnetic field recordings, leading to larger than expected errors in the data. The authors should comment accordingly in the rewritten paper.

4. In line 102 the authors mention the Keetmanshoop INTERMAGNET observatory. Please provide a reference. (Korte, M., M. Mandea, H.-J. Linthe, A. Hemshorn, P. Kotzé and E. Ricaldi : New geomagnetic field observations in the South Atlantic Anomaly region. *Annals of Geophysics*, 52, 65-81, 2009.)

5. In line 106 the authors briefly mention the use of venting pipes for temperature stability. It is well-known that fluxgate magnetometers are extremely sensitive to temperature variations. Are venting pipes adequate to provide the required temperature stability? It would be informative for the reader to add a temperature variation plot to show that the environment inside the box is stable enough for the fluxgate sensor.

6. What about lightning protection? The authors do not mention it in the paper, and is it of any concern?

7. The authors mention that data are sampled at 1 min intervals using 1 sec values. How is this determined? Using an average over 60 sec, centred at the middle of each

minute interval, or is it done by taking the average 30sec before and 30 sec after the minute? Please explain.

8. Line 330: Please provide a website if no journal reference is available.

---

## Author Comment (AC2)

Dr. A. Nel
*SANSA Space Science*
*Hospital Road 32, Hermanus*
*South Africa*
✉ anel@sansa.org.za

**Response to reviewers** September 10, 2025

We would like to thank the Reviewers for their constructive and thoughtful comments. We have addressed each point and revised the manuscript accordingly. Our detailed responses follow below.

**Reviewer 1**

**Comment 1**

**The 'Entoto Observatory' can at best be called a 'Variometer Station' because it does not qualify as a magnetic observatory in the true sense of the word as no absolute measurements are being done to determine baseline values. In the manuscript the authors use both 'station' and 'observatory' which can confuse the reader of this paper**

We thank the reviewer and agree that the station should not be referred to as an observatory. We now refer to it consistently as the Entoto Magnetometer Station and have clarified in the abstract and body that it is a variometer-only station hosted at the Entoto Observatory site, without absolute baseline measurements.

**Comment 2**

**In line 7 the authors write 'ENTOTO', while in the title of the manuscript it is written 'Entoto'. Consistency is advised.**

We agree that consistency is important, and we have revised the manuscript accordingly to use the capitalisation "Entoto" throughout.

**Comment 3**

**Figure 1 reveals that substantial magnetic gradients exist at the site of the station, reaching 240 nT/m in the North-West corner. The effect of this is that the area is not clean as a 10 nT/m gradient is normally the criterium for a clean site. It is recommended that the position of the station be shown in Figure 1 to give the reader a better appreciation of its location. These large gradients, indicative of magnetised rocks under the surface, can unfortunately lead to substantial induction effects on the magnetic field recordings, leading to larger than expected errors in the data. The authors should comment accordingly in the rewritten**

**paper.**

We appreciate this observation. We have now added the station's precise position to Figure 1 and discussed the implications of high magnetic gradients. Although the gradient reaches 240 nT/m in the NW corner, the selected deployment site lies outside this region. As this is not an absolute observatory, we agree that such gradients could introduce induction effects but are acceptable given the station's purpose for space weather monitoring. A note on this limitation has been added to the discussion section.

━━━━━ Comment 4

**In line 102 the authors mention the Keetmanshoop INTERMAGNET observatory.Please provide a reference. (Korte, M., M. Mandea, H.-J. Linthe, A. Hemshorn, P.Kotzé and E. Ricaldi : New geomagnetic field observations in the South Atlantic Anomaly region. Annals of Geophysics, 52, 65-81, 2009.)**

We thank the reviewer for this helpful reference suggestion. We have included the citation as recommended to support the mention of the Keetmanshoop INTERMAGNET observatory. The reference will be added to the bibliography and cited appropriately in the main text.

━━━━━ Comment 5

**In line 106 the authors briefly mention the use of venting pipes for temperature stability. It is well-known that fluxgate magnetometers are extremely sensitive to temperature variations. Are venting pipes adequate to provide the required temperature stability? It would be informative for the reader to add a temperature variation plot to show that the environment inside the box is stable enough for the fluxgate sensor.**

We thank the reviewer for raising this important point. We agree that venting pipes alone cannot guarantee the required thermal stability for fluxgate magnetometers. To assess and address this, we analysed the co-located sensor (T1) and electronics (T2) temperatures. As shown in Figure 3, the enclosure moderates the thermal environment, with modest daily variations (median ∼3.5°C for T1 and ∼4.6°C for T2; Table 1). This corresponds to an estimated median peak-to-peak effect of only ∼2.6 nT on the H component.

To further mitigate residual effects, we derived and applied a temperature coefficient of –0.733 nT °C$^{-1}$ for H. Figure 4 demonstrates that the correction slightly reduces the small diurnal variation in $\Delta F$ (on the order of ∼2 nT on quiet days) and lowers its correlation with temperature. The effect is subtle, but consistent with expectations and shows that the correction acts as a safeguard. Overall, while the Entoto enclosure does not achieve observatory-grade stability, the combination of moderated thermal conditions and post-processing correction is adequate for variometer-grade operation.

━━━━━ Comment 6

**What about lightning protection? The authors do not mention it in the paper, and is it of any concern?**

At present, the Entoto Magnetometer Station does not have a dedicated lightning protection system installed. The sensor cable is not laid directly on the ground, which reduces the likelihood of induced currents reaching the sensor in the event of a lightning strike. The most vulnerable point is the AC power supply line, which could be affected by a direct hit; however, due to the isolated sensor grounding and layout, we do not expect significant risk of damage to the fluxgate sensor itself.

We have now noted in the revised manuscript that while no dedicated system is yet installed, lightning protection measures are under active consideration for future upgrades.

**Comment 7**

**The authors mention that data are sampled at 1 min intervals using 1 sec values. How is this determined? Using an average over 60 sec, centred at the middle of eachminute interval, or is it done by taking the average 30sec before and 30 sec after the minute? Please explain.**

We thank the reviewer for the observation. The 1-minute values in our analysis are computed as centered means from the 1-second data, following the standard method recommended by IAGA and INTERMAGNET. We will clarify this in the manuscript accordingly.

**Comment 8**

**Line 330: Please provide a website if no journal reference is available.**

The citation in line 330 was an incomplete reference to the CHAOS-8.2 geomagnetic field model. We will correct this and include the full journal reference in the bibliography.

**Reviewer 2**

**Comment 1**

**Line 48: Note INTERMAGNET does not operate observatories but rather enables exchange of data. The stations mentioned are (or were) operated by IPGP (AAE, MBO, SOK) and IPGP and CRAAG (TAM)**

We thank the reviewer for this correction. We will revise the sentence to correctly attribute the operation of the observatories to IPGP and CRAAG, and clarify that INTERMAGNET facilitates global data exchange rather than operating observatories directly.

**Comment 2**

**Figure 1. Error in caption. Longitude should read...**

Corrected as suggested.

**Comment 3**

Figure 1 suggests high magnetic gradients (up to 20 nT/m) compared to the accepted recommendation of 1 nT/m for a typical geomagnetic observatory (Jankowski and Sucksdorff, 1996). However, that recommendation is typically applied to absolute magnetic observatories, so may not be significant if this observatory is primarily designed for space weather monitoring. Can this be commented on, particularly given the statement in Line 90 on static disturbances?

We now clarify that the 1 nT/m criterion (Jankowski and Sucksdorff, 1996) applies to absolute observatories. As Entoto is a variometer station intended for space weather research, higher gradients, while not ideal, are acceptable. This is now stated in both Figure 1 and the main text.

**Comment 4**

**Line 75 The definition of L9 is made further on in the paper but can it be included here as 'lower K = 9 limit'?**

We have moved the definition of L9 and now include "L9: the lower K = 9 threshold value" at its first mention in the text.

**Comment 5**

**Line 58,84 Can these statements on the importance of an East African, equatorial station be strengthened by citation to literature on the requirement for regional space weather monitoring?**

We have strengthened this with references to: - Hamid et al. (2014) - Myint et al. (2022) - Mungufeni et al. (2018)

These support the importance of equatorial stations for monitoring EEJ and regional space weather.

**Comment 6**

**Line 113 Does the specification here of '1-second resolution' refer to direction or time? Could this be clarified?**

Thank you for the observation. It refers to temporal resolution. We have updated the sentence to clarify this.

**Comment 7**

**Line 124 The term 'geomagnetic coordinate system' would be more accurately described as 'geodetic coordinate system' or 'geographic coordinate system' given the definition of the XYZ co-ordinate system**

We have corrected this to "geographic coordinate system".

**Comment 8**

**Line 131 Is the fact that the data are sampled at one-minute intervals contradictory to the 1-second resolution referred to in Line 124? Are the data down sampled or filtered to one-minute? If so, can this process be defined i.e. is a specific filter used?**

We clarified that no digital filtering is applied; 1-minute values are computed as simple centered means of 60 consecutive 1-second samples.

**Comment 9**

**Line 172 What are the sources of non-geomagnetic noise in this frequency band (¿ 2-minute period) and can these be filtered without attenuating the signal of interest in the same band?**

Non-geomagnetic noise sources include temperature drift and anthropogenic EM interference. While some are corrected (e.g., temperature), others cannot be removed without potentially attenuating ionospheric signals. We have added a note on these trade-offs.

**Comment 10**

**Figures 9 and 10 The fitting of the mean daily maximum EEJ amplitude in the figures is close to the median daily maximum. If the daily maximum is normally distributed for both quiet and storm conditions, then the mean and the median will, of course, be equal but can it be confirmed that the plots show the mean and not the median?**

These values are confirmed to be the mean daily maxima, as implemented in our processing scripts (Appendix).

**Comment 11**

**Given the noted operational difficulties in maintaining long-term magnetic stations in the region, can the authors comment further on measures taken to ensure the ENTOTO station will continue to operate in the long-term? For example, are there formal long-term agreements in place between the institutes collaborating on this project?**

Thank you for raising this important point. We have added a note in the manuscript outlining the sustainability strategy for the Entoto station.

The deployment is supported by a formal Memorandum of Understanding (MoU) between the South African National Space Agency (SANSA) and the Space Science and Geospatial Institute (SSGI) in Ethiopia. This framework ensures long-term institutional commitment, while also facilitating data sharing, infrastructure access, and joint research efforts.

We would like to acknowledge and credit earlier initiatives, particularly the efforts of teams such as the AMBER group, whose deployments across Africa laid critical groundwork. From their and other teams' experiences, we've learned the importance of maintaining continuous and open communication between hosting and supporting institutes. This includes regular updates between technicians, researchers, and software teams to avoid the "black box" problem often encountered in remote or distributed sensor networks.

Our approach prioritises mutual scientific partnership, with cross-training on instrument maintenance, data processing, and analysis. The aim is to support the SSGI to one day be the second geomagnetic working group on the continent, who in turn can support SANSA in its endeavours. This ensures operational resilience and builds local expertise and longevity of instrument networks. We also continue to pursue travel and capacity-building grants to support technical exchanges and joint workshops, reinforcing both technical continuity and regional collaboration.

**Appendix: EEJDSTv4.py, Python Script for EEJ Signal Processing**

```python
import os
import pandas as pd
import numpy as np
import chaosmagpy as cp
import requests
import matplotlib.pyplot as plt
from datetime import datetime, timedelta
from sklearn.linear_model import LinearRegression
import importlib
import calcChaos
importlib.reload(calcChaos)
from calcChaos import chaos, chaos_ext, datetime_to_decimal_year
import re
from datetime import datetime
from tqdm import tqdm
import matplotlib.dates as mdates def load_entoto_data(directory):
    all_data = []

    def extract_date(filename):
        match = re.search(r'ent(\d{8})pmin\.min', filename)
        if match:
            return datetime.strptime(match.group(1), '%Y%m%d')
        return datetime.min  # fallback

    # List and sort .min files by date in filename
    min_files = sorted(
        [f for f in os.listdir(directory) if f.endswith('.min')],
        key=extract_date
    )

    # Progress bar over files
    for file in tqdm(min_files, desc="Loading Entoto .min files"):
        file_path = os.path.join(directory, file)
        try:
            df = pd.read_csv(
                file_path,
                sep=r'\s+',
                comment='#',
                header=None,
```

```
    skiprows=16,
    names=["DATE", "TIME", "DOY", "ENTX", "ENTY", "ENTZ", "ENTF"],
    engine='python',
    on_bad_lines='skip'
)
df.replace(99999.0, np.nan, inplace=True)
# Combine DATE and TIME into a single DATETIME column
df['DATETIME'] = pd.to_datetime(df['DATE'] + ' ' + df['TIME'],
                errors='coerce')
# Drop the original separate DATE and TIME columns
df.drop(columns=['DATE', 'TIME'], inplace=True)
all_data.append(df)
except Exception as e:
print(f"Error processing {file}: {e}")
return pd.concat(all_data, ignore_index=True) if all_data else None
# Step 2: Calculate H component from X and Y
def calculate_H_component(df):
df['H'] = np.sqrt(df['ENTX']**2 + df['ENTY']**2)
return df
# Step 3: Remove CHAOS internal field to get H_residual
def remove_internal_field(df, station_lat=9.108, station_lon=38.807,
        station_alt=2450):
df['H_internal'] = np.nan
df['H_residual'] = np.nan
# Group data by date
df['DATE'] = df['DATETIME'].dt.date
unique_dates = df['DATE'].unique()
for date in unique_dates:
daily_df = df[df['DATE'] == date]
pkl_filename = f"internal_field_{date}.pkl"
if os.path.exists(pkl_filename):
    daily_internal = pd.read_pickle(pkl_filename)
else:
    # Compute internal field for each timestamp
    internal_values = []
    for dt in daily_df['DATETIME']:
        Bx, By, Bz = chaos(datetime_to_decimal_year(dt), station_lat,
                        station_lon, station_alt)
        H_internal = np.sqrt(Bx**2 + By**2)
        internal_values.append(H_internal)
    daily_internal = pd.Series(internal_values, index=daily_df.index)
    daily_internal.to_pickle(pkl_filename)
df.loc[daily_df.index, 'H_internal'] = daily_internal
df.loc[daily_df.index, 'H_residual'] = df.loc[daily_df.index, 'H'] -
                daily_internal
```

```python
df.drop(columns=['DATE'], inplace=True)
return df
def compute_external_field(df, station_lat=9.108, station_lon=38.807,
           station_alt=2450):
df['H_external'] = np.nan
# Group data by date
df['DATE'] = df['DATETIME'].dt.date
unique_dates = df['DATE'].unique()
for date in unique_dates:
   daily_df = df[df['DATE'] == date]
   pkl_filename = f"external_field_{date}.pkl"
   if os.path.exists(pkl_filename):
       daily_external = pd.read_pickle(pkl_filename)
   else:
       # Compute external field for each timestamp
       external_values = []
       for dt in daily_df['DATETIME']:
           Bx_ext, By_ext, Bz_ext = chaos_ext(datetime_to_decimal_year(dt)
                           , station_lat, station_lon, station_alt)
           H_external = np.sqrt(Bx_ext**2 + By_ext**2)
           external_values.append(H_external)
       daily_external = pd.Series(external_values, index=daily_df.index)
       daily_external.to_pickle(pkl_filename)
   df.loc[daily_df.index, 'H_external'] = daily_external
df.drop(columns=['DATE'], inplace=True)
return df
# Step 4: Estimate average night-time magnetospheric field from H_residual
def estimate_magnetospheric_component(df):
night_mask = (df['DATETIME'].dt.hour >= 18) | (df['DATETIME'].dt.hour < 6)
df['H_magnetospheric'] = df.loc[night_mask, 'H_residual']
return df
# Step 5: Extract daytime EEJ signal from H_residual (still contains
           magnetospheric field)
def extract_eej_signal(df):
day_mask = (df['DATETIME'].dt.hour >= 9) & (df['DATETIME'].dt.hour <= 15)
df['EEJ'] = df.loc[day_mask, 'H_residual']
return df
def fetch_dst_index(start_date, end_date):
dst_records = []
filepath="/home/amore/Documents/00Data/Dst_oct2024_apr2025.dat"
with open(filepath, 'r') as file:
   for line in file:
       parts = line.strip().split()
       if len(parts) < 26:
           continue  # Skip malformed lines
```

```python
           # Parse date from ID like DST2410*01PPX120
           id_str = parts[0]
           year = int("20" + id_str[3:5])
           month = int(id_str[5:7])
           day = int(id_str.split("*")[1][:2])
           try:
               hourly_values = [int(val) for val in parts[2:26]]  # Skip 2nd
                                column (always 0), then 24 values
           except ValueError:
               continue  # Skip lines with invalid integer entries
           for hour, dst in enumerate(hourly_values):
               dt = datetime(year, month, day, hour)
               if start_date <= dt <= end_date:
                   dst_records.append({'DATETIME': dt, 'Dst': dst})
return pd.DataFrame(dst_records)
# Step 7: Perform linear regression between Dst and H_residual to estimate
        magnetospheric field
# Step 8: Subtract modeled magnetospheric contribution to get cleaned EEJ
        signal (HEEJ)
def perform_linear_regression(df, dst_data):
# Ensure both are sorted by time
df = df.sort_values('DATETIME')
dst_data = dst_data.sort_values('DATETIME')
# Merge with nearest previous Dst value (i.e., forward fill)
merged = pd.merge_asof(df, dst_data[['DATETIME', 'Dst']], on='DATETIME',
            direction='backward')
# Drop rows with missing data
merged_clean = merged.dropna(subset=['Dst', 'H_residual'])
# Prepare regression inputs
x = merged_clean['Dst'].values.reshape(-1, 1)
y = merged_clean['H_residual'].values.reshape(-1, 1)
# Perform linear regression
reg = LinearRegression().fit(x, y)
merged_clean['Hmag_model'] = reg.predict(x)
# Merge the modeled magnetospheric signal back into the full dataset
merged = pd.merge(merged, merged_clean[['DATETIME', 'Hmag_model']], on='
            DATETIME', how='left')
# Compute the cleaned EEJ signal
merged['HEEJ'] = merged['H_residual'] - merged['Hmag_model']
return merged, reg.coef_[0][0]
# After regression, keep only daytime values of the cleaned signal:
def extract_daytime_eej(df):
df = df.copy()  # prevent SettingWithCopyWarning
```

```python
df['HEEJ_daytime'] = np.nan   # initialize the column with NaNs
day_mask = (df['DATETIME'].dt.hour >= 9) & (df['DATETIME'].dt.hour <= 15)
df.loc[day_mask, 'HEEJ_daytime'] = df.loc[day_mask, 'HEEJ']
return df
def plot_monthly_magnetospheric_vs_dst(df, dst_data):
df['YEAR'] = df['DATETIME'].dt.year
df['MONTH'] = df['DATETIME'].dt.month
for (year, month), group in df.groupby(['YEAR', 'MONTH']):
  start = group['DATETIME'].min()
  end = group['DATETIME'].max()
  dst_subset = dst_data[(dst_data['DATETIME'] >= start) & (dst_data['
                   DATETIME'] <= end)]
  plt.figure(figsize=(12, 6))
  ax1 = plt.gca()
  ax1.plot(group['DATETIME'], group['H_magnetospheric'], label='
                   H_magnetospheric', color='blue')
  ax1.set_ylabel('H_magnetospheric (nT)', color='blue')
  ax1.tick_params(axis='y', labelcolor='blue')
  ax2 = ax1.twinx()
  ax2.plot(dst_subset['DATETIME'], dst_subset['Dst'], label='Dst Index',
                   color='red')
  ax2.set_ylabel('Dst Index (nT)', color='red')
  ax2.tick_params(axis='y', labelcolor='red')
  plt.title(f'Magnetospheric Signal vs Dst - {year}-{month:02}')
  ax1.set_xlabel('Date')
  plt.grid()
  plt.tight_layout()
  plt.savefig(f'Magnetospheric_vs_Dst_{year}_{month:02}.png', dpi=300)
  plt.close()
# Step 10: Plot raw EEJ signal (before Dst correction)
def plot_superposed_epoch_eej(df):
if 'EEJ' not in df.columns:
  print("EEJ not available. Skipping superposed epoch plots.")
  return
df['YEAR'] = df['DATETIME'].dt.year
df['MONTH'] = df['DATETIME'].dt.month
df['HOUR_MIN'] = df['DATETIME'].dt.strftime('%H:%M')
# Keep only daytime
df_daytime = df[(df['DATETIME'].dt.hour >= 9) & (df['DATETIME'].dt.hour <=
                   15)].copy()
# Round time to 30-minute bins
df_daytime['DATETIME'] = df_daytime['DATETIME'].dt.floor('30T')
df_daytime['HOUR_MIN'] = df_daytime['DATETIME'].dt.strftime('%H:%M')
for (year, month), group in df_daytime.groupby(['YEAR', 'MONTH']):
```

```python
# Pivot: time of day (rows)     day (columns)
pivot = group.pivot_table(index='HOUR_MIN', columns=group['DATETIME'].
                dt.date, values='EEJ')
# Compute mean and std at each time bin
mean_series = pivot.mean(axis=1)
std_series = pivot.std(axis=1)
# Convert HOUR_MIN back to datetime-like index for proper plotting
time_labels = [datetime.strptime(t, '%H:%M') for t in mean_series.index
                ]
plt.figure(figsize=(10, 5))
plt.plot(time_labels, mean_series, label='Mean EEJ', color='orange')
plt.fill_between(time_labels, mean_series - std_series, mean_series +
                std_series,
                color='orange', alpha=0.3, label=' 1  Std Dev')
# Format x-axis to show only time (HH:MM)
import matplotlib.dates as mdates
plt.gca().xaxis.set_major_formatter(mdates.DateFormatter('%H:%M'))
plt.xlabel('Local Time (24-hour)')
plt.ylabel('EEJ Magnetic Field (nT)')
plt.title(f'Superposed Epoch of EEJ - {year}-{month:02}')
plt.grid()
plt.legend()
filename = f'Superposed_EEJ_{year}_{month:02}.png'
plt.savefig(filename, dpi=300)
plt.close()
def plot_superposed_epoch_eej_vs_heej(df):
if 'EEJ' not in df.columns or 'HEEJ' not in df.columns:
print("EEJ or HEEJ not available. Skipping superposed comparison plots.
                ")
return
df['YEAR'] = df['DATETIME'].dt.year
df['MONTH'] = df['DATETIME'].dt.month
df['HOUR_MIN'] = df['DATETIME'].dt.strftime('%H:%M')
# Filter for daytime hours
df_daytime = df[(df['DATETIME'].dt.hour >= 9) & (df['DATETIME'].dt.hour <=
            15)].copy()
df_daytime['DATETIME'] = df_daytime['DATETIME'].dt.floor('30min')
df_daytime['HOUR_MIN'] = df_daytime['DATETIME'].dt.strftime('%H:%M')
for (year, month), group in df_daytime.groupby(['YEAR', 'MONTH']):
# Pivot tables for EEJ and HEEJ
pivot_eej = group.pivot_table(index='HOUR_MIN', columns=group['DATETIME
                '].dt.date, values='EEJ')
pivot_heej = group.pivot_table(index='HOUR_MIN', columns=group['
                DATETIME'].dt.date, values='HEEJ')
# Compute mean values
```

```
mean_eej = pivot_eej.mean(axis=1)
mean_heej = pivot_heej.mean(axis=1)
# Create time axis
time_labels = [datetime.strptime(t, '%H:%M') for t in mean_eej.index]
plt.figure(figsize=(10, 5))
plt.plot(time_labels, mean_eej, label='Raw␣EEJ', color='orange')
plt.plot(time_labels, mean_heej, label='Dst-corrected␣EEJ␣(HEEJ)',
              color='green')
plt.gca().xaxis.set_major_formatter(mdates.DateFormatter('%H:%M'))
plt.xlabel('Local␣Time␣(24-hour)')
plt.ylabel('Magnetic␣Field␣(nT)')
plt.title(f'Superposed␣Epoch␣of␣EEJ␣vs␣HEEJ␣-␣{year}-{month:02}')
plt.grid()
plt.legend()
plt.tight_layout()
filename = f'Superposed_EEJ_vs_HEEJ_{year}_{month:02}.png'
plt.savefig(filename, dpi=300)
plt.close()
def filter_by_month(df, year, month):
# Ensure DATETIME is datetime type
df['DATETIME'] = pd.to_datetime(df['DATETIME'])
# Filter for the specific year and month
filtered_df = df[(df['DATETIME'].dt.year == year) & (df['DATETIME'].dt.
          month == month)]
# Filter for times between 09:00 and 15:00
filtered_df = filtered_df[(filtered_df['DATETIME'].dt.hour >= 9) & (
          filtered_df['DATETIME'].dt.hour < 15)]
print(filtered_df)
# Main execution pipeline
def main():
directory = '/home/amore/Documents/00Data/ENT0'
output_file = 'testfile.pkl'
df = load_entoto_data(directory)
if df is None:
print("No␣data␣files␣found.")
return
df = calculate_H_component(df)
df = remove_internal_field(df)
df = compute_external_field(df)
df = extract_eej_signal(df)
```

```
df = estimate_magnetospheric_component(df)
plot_superposed_epoch_eej(df)
start_date, end_date = df['DATETIME'].min(), df['DATETIME'].max()
dst_data = fetch_dst_index(start_date, end_date)
if dst_data is not None and not dst_data.empty:
  df, k = perform_linear_regression(df, dst_data)
  print(f"Estimated scaling factor k: {k:.3f}")
  df = extract_daytime_eej(df)
  filter_by_month(df, 2025,3)
  plot_monthly_magnetospheric_vs_dst(df, dst_data)
  plot_superposed_epoch_eej_vs_heej(df)
else:
  print("Warning: No Dst data available for the date range. Skipping
                   regression and EEJ correction plot.")
df.to_pickle(output_file)
print(f"Processed data saved to {output_file}")
```

Listing 1: Python script used to compute and aggregate EEJ amplitude from Entoto station data.

---

## Author Comment (AC3)

Geoscientific Instrumentation, Methods and Data Systems (gi)

**First results from the equatorial geomagnetic station at Entoto Observatory and Research Center**

Amoré Nel[1,2], Nigussie Giday[3], Marcos Da Silva[4], Daniel Chekole[3], Jürgen Matzka[4], Ziyaad Isaacs[1], Oliver Bronkala[4], and Lamessa Mogasa[3]

[1]South African National Space Agency (SANSA), South Africa.
[2]Center for Space Research, North-West University, Potchefstroom, 2522, South Africa.
[3]Department of Space and Planetary Science, Space Science and Geospatial Institute (SSGI), Addis Ababa, Ethiopia.
[4]GFZ German Research Centre for Geosciences, Potsdam, Germany.

**Correspondence:** Amoré Nel (anel@sansa.org.za)

[revised manuscript text omitted]

The scarcity of operational equatorial magnetometer stations in Africa presents a major obstacle to advancing geomagnetic and space weather research in the region (Hamid et al., 2014a; Myint et al., 2022a; Mungufeni et al., 2018). Without high-resolution, continuous ground-based observations, researchers must rely on satellite data, which, while valuable, lacks the temporal resolution necessary for detailed EEJ and Sq variation analysis. Additionally, the absence of equatorial ground-based measurements limits the ability to validate global geomagnetic models and weakens Africa's contribution to international space weather monitoring efforts.

The establishment of the Entoto Magnetometer Station represents a significant step toward addressing this gap. By providing high-frequency geomagnetic measurements from a location near the magnetic equator, the station enables in-depth investigations of the EEJ, Sq variations, and their interactions with global and regional space weather phenomena. The station's data will contribute to both regional forecasting and global geomagnetic modeling, enhancing our understanding of equatorial electro-

[revised manuscript text omitted]
 < -50) times respectively. These Figures illustrate the ability of the Entoto station to capture both magnetospheric and ionospheric responses to space weather conditions. Figure 10 demonstrates a moderate to strong Pearson correlation of 0.62 between the station's derived magnetospheric signal and the global Dst index, indicating that the station reliably reflects variations in magnetospheric current systems. Figures 11 and 12 compare the equatorial electrojet (EEJ) signal during geomagnetically quiet and disturbed periods. The mean daily maximum EEJ amplitude was significantly larger during storm days ($-317.9$ nT) than during quiet days ($-257.2$ nT), suggesting that the station effectively captures enhancements in the EEJ associated with storm-time ionospheric dynamics. These preliminary results confirm the station's sensitivity to both global geomagnetic disturbances and regional ionospheric variability.

[revised manuscript text omitted]